# Comparative Study of the Properties of Wood Flour and Wood Pellets Manufactured from Secondary Processing Mill Residues

**DOI:** 10.3390/polym13152487

**Published:** 2021-07-28

**Authors:** Geeta Pokhrel, Yousoo Han, Douglas J. Gardner

**Affiliations:** 1School of Forest Resources, 5755 Nutting Hall, University of Maine, Orono, ME 04469, USA; yousoo.han@maine.edu; 2Advanced Structures and Composites Center, University of Maine, 35 Flagstaff Road, Orono, ME 04469, USA

**Keywords:** secondary processing mill residues, wood flour, wood pellets, wood–plastic composites, physical properties, transportation costs

## Abstract

The generation of secondary processing mill residues from wood processing facilities is extensive in the United States. Wood flour can be manufactured utilizing these residues and an important application of wood flour is as a filler in the wood–plastic composites (WPCs). Scientific research on wood flour production from mill residues is limited. One of the greatest costs involved in the supply chain of WPCs manufacturing is the transportation cost. Wood flour, constrained by low bulk densities, is commonly transported by truck trailers without attaining allowable weight limits. Because of this, shipping costs often exceed the material costs, consequently increasing raw material costs for WPC manufacturers and the price of finished products. A bulk density study of wood flour (190–220 kg/m^3^) and wood pellets (700–750 kg/m^3^) shows that a tractor-trailer can carry more than three times the weight of pellets compared to flour. Thus, this study focuses on exploring the utilization of mill residues from four wood species in Maine to produce raw materials for manufacturing WPCs. Two types of raw materials for the manufacture of WPCs, i.e., wood flour and wood pellets, were produced and a study of their properties was performed. At the species level, red maple 40-mesh wood flour had the highest bulk density and lowest moisture content. Spruce-fir wood flour particles were the finest (d_gw_ of 0.18 mm). For all species, the 18–40 wood flour mesh size possessed the highest aspect ratio. Similarly, on average, wood pellets manufactured from 40-mesh particles had a lower moisture content, higher bulk density, and better durability than the pellets from unsieved wood flour. Red maple pellets had the lowest moisture content (0.12%) and the highest bulk density (738 kg/m^3^). The results concluded that the processing of residues into wood flour and then into pellets reduced the moisture content by 76.8% and increased the bulk density by 747%. These material property parameters are an important attempt to provide information that can facilitate the more cost-efficient transport of wood residue feedstocks over longer distances.

## 1. Introduction

Many forest-based industries, including sawmills and turning mills, produce residues often termed “secondary processing mill residues”. Secondary processing mill residues are the remnants after timber has been utilized in the wood mills for the processing of primary products such as lumber. Secondary processing residues are usually clean, uniform, on-site and low in moisture content and can be prepared for further utilization. Some of the examples include sawdust, planar shavings, small chips, etc. They have several direct or indirect applications such as the production of heat and energy, as the raw material for particleboard, pellets or char, for agricultural use (livestock bedding), and landscape applications. In the US, the volume of wood residues generated in 2019 was 15,288,915 m^3^ and the volume of wood chips and particles was 46,355,003 m^3^ [1]. The state of Maine generates around 1.6 million tons of green residues from the wood mills each year. Maine produces roughly 800 million board feet, from which roughly 800, 000 tons of clean chips and 800,000 tons of bark and sawdust are produced [2]. Kingsley [3] has mentioned, as a decent rule of thumb, that for every 1000 board feet of lumber produced, 2 tons of residues are produced, out of which 1 ton is clean chips (free from bark, adhesives, metals, microorganism attack, etc.).

One of the applications of mill residues can be grinding them into a powder form referred to as “wood flour” or “wood fiber”. Wood flour is a finely ground wood cellulose. It is a highly comminuted wood material with a very fine particle size distribution typically produced from mill residues using several grinder types and sized by mechanical or air screening methods [4]. Wood flour is composed of fine particles passing through a screen with 850-micron openings or 20 US standard mesh [5]. Such reduced particle sizes can be achieved by fine grinders (e.g., hammermill, disc refiner, pin mill or chain mill). The reduction in biomass size changes the particle size and shape, increases bulk density, improves flow properties, increases porosity, and generates new surface area [6,7]. Compared to the other grinding mills, hammer mills have an advantage because of their ability to finely grind a greater variety of materials [8]. Nominal biomass particle sizes produced by hammer mill grinding depend on the processing characteristics of the mill. The two material variables that distinguish wood flour are the species and size [9]. High-quality wood flour can be made from hardwoods attributable to their durability and strength. Commercial production of wood flour started in 1906, and the first commercial product from wood flour was a gear shift knob for a Rolls Royce automobile in 1916 [10]. Wood flour uses can be grouped into absorbent, chemically reactive substances, chemically inert fillers, modifiers of physical properties, mildly abrasive materials, and decorative materials. Species mostly preferred for wood flour production include white pine (eastern, western and sugar pines), aspen, spruce, hemlock and, to some extent, balsam fir, paper birch, and the southern pines. In the US, 75% of wood flour is manufactured from white pine. The greatest application of wood flour is its utilization as lignocellulosic fiber for reinforcing plastics. Wood filler is compounded into the polymer matrix in order to enhance the properties of the polymer to behave more similarly to wood than a polymer. Wood–plastic composites (WPCs) have significant advantages from material, environmental, and economic perspectives [11,12,13].

Similarly, the other application of mill residues can be as a source of raw material for manufacturing pellets. Sources of raw materials for wood pellets include fallen branches, thinning and broken stems from the forest, and residues generated in sawmills such as sawdust, chips, etc. Wood pellets are 0.5–3 cm long and cylindrical (6–8 mm) compressed materials produced in pellet mills under high pressure (≈300 MPa) and high temperature (≈120 °C) [14]. Compression of the biomass or residues into pellets involves elastic and plastic deformation of particles and softening of natural binders such as starch, protein, lignin, fats and fibers for binding the particles together [15]. The type and amounts of extractives contribute to the fundamental difference between the softwood and hardwood pellets [16]. According to Calderon et al. [17], the utilization of wood pellets throughout the world has increased from approximately 12 million metric tons in 2008 to 56 million metric tons in 2018.

Transportation of wood flour is conducted using multi-walled paper bags (approximately 23 kg or 50 lbs) or bulk bags (typically 1.5 cubic meters or 55 cubic feet) or by bulk trailers [6]. Wood flour, being a low-density fluffy material, occupies less bulk weight for transportation. Considering shipment to the point of destination, the cost is often more than that for sawdust [5]. Wood flour transportation over longer distances can incur excess shipping costs as compared to the material price. Compared to other biomass fuels, pellets are easy to handle, store and transport [18]. Wood pellets, compared to other fibrous materials of wood and solid biofuel materials, have an increased energy output per unit volume [19]. The average bulk density of wood flour is 190–220 kg/m^3^ or 12–14 lb./ft^3^ [20] and the bulk density of wood pellets is 700–750 kg/m^3^ or 43–47 lb./ft^3^ [21]. This comparative study shows that the average bulk density of wood pellets is four times greater than wood flour, which suggests the storage footprint area for wood pellets is reduced by four times.

Very little information is available in the literature and technical sources regarding the secondary processing mill residues and their applications. On the other hand, there is no recent research on wood flour production from mill residues and its properties’ characterization. Studies on wood flour production are quite outdated [5,21]. This suggests there is a wide gap to find current research on wood flour production. Nevertheless, a comparative study of the properties of wood flour based on different wood species and various mesh size level is still lacking. Literature regarding the applications of mm dimension residues on the pelletization of wood pellets is abundant. However, fine micron scale wood fibers being processed into pellets is limited in literature sources. There has been considerable research on WPC manufacturing using wood flour and studies on its properties, but only limited work has been reported on WPC manufacturing using wood pellets. Butylina et al. [22] reported on the comparative properties of WPCs using wood flour, wood pellets and heat-treated fibers in a polypropylene matrix. This study is focused on comparing wood flour and wood pellets with the overall goal of reducing the transportation costs of wood fillers for WPC manufacturing.

In addressing all the above-mentioned motives, one of the objectives of this study is to produce wood flour using the clean mill residues in Maine and characterize its properties. The characterization, i.e., the study of the morphology, moisture content, bulk density, particle size distribution and aspect ratio of the wood flour from each of the four Maine species (Northern White Cedar, Eastern White Pine, Spruce-Fir and Red Maple) can help better understand their specific properties. The second objective is the application of fine wood flour from each wood species in the manufacturing of wood pellets, and the studying of its properties. Finally, a comparative analysis of the mill residues, flour and pellets, focusing mostly on the parameters that affect production and transportation, i.e., in terms of moisture and density, is carried out. This study hopes to convey how each processing step causes a change in these material parameters. We expect this study will be a baseline study for future work on wood and polymer composite manufacturing from different raw materials in an attempt to gain maximum manufacturing efficiency.

## 2. Materials and Methods

### 2.1. Materials

Mill residues of around 100–150 kg typical planar shavings, sawdust, and small chips were obtained from local wood mills in Maine. Northern White Cedar (*Thuja occidentalis*) from (Katahdin Forest Products, Oakfield, ME, USA), Eastern White Pine (*Pinus strobus*) from (Hancock Lumber, Pittsfield, ME, USA), Eastern Spruce-Balsam Fir (*Picea rubens-Abies balsamea*) from (Pleasant River Lumber, Dover-Foxcroft, ME, USA) and Red Maple (*Acer rubrum*) from (Lumbra Hardwoods, Milo, ME, USA) were obtained through on-site visits. Figure 1 below presents the images of secondary processing processing mill residues of four different wood species in the study area. The scale bar is 3 cm in length.These residues were clean and free from bark, adhesives, metals, etc., and were either air-dried or kiln-dried. These residues were utilized to manufacture wood flour and then wood pellets. The raw material for manufacturing wood pellets, i.e., wood flour, was produced in the lab.

### 2.2. Manufacturing Process of Wood Flour and Wood Pellets

A Bliss Eliminator Hammermill (Bliss Industries LLC, Ponca City, OK, USA) with a screen size of 0.5 mm was used to produce wood flour. The residues were manually fed into the hopper of the hammermill separately for each species. The faces, edges, and corners of the hammers cut and shattered the material and threw it forcibly against the casing. Further size reduction took place in the layers of the material retained on the screen. The wood flour was then removed from the collection box. A Gilson screens haker (Gilson Company Inc., Worthington, OH, USA) was then used for screening the wood flour fractions into 20, 40, 60, 80, and 100 mesh sizes to study different characteristics of the wood flour, including morphology and other physical properties.

Usually, for the manufacturing of wood pellets that have other applications, for example, as biofuels, raw materials of up to 2 mm in size are preferred. However, in our experiments, the particle sizes were smaller (<500 µm). Several literature sources describe material characteristics, processing methods, and production parameters for wood pellet production [23,24,25]. For each species, two categories of wood pellets, one with unsieved flour and the other with a 40 mesh size fraction, were used for the manufacturing of pellets. The wood pellets were pelletized at the Technology Research Center (TRC), Old Town, ME, USA. A Lawson Mills Pellet Mill LM72A (Lawson Mills Biomass Solutions Ltd., North Wiltshire, PE, Canada) was used for the production of wood pellets. An integrated fines removal system captured fines for recycling. This machine allows up to two additives for binding. Even though the capacity of the pelletizer depends on the raw material, this machine processes as much as 160 kg of materials per hour. The ground wood flour from the hammer mill had a relatively low moisture content as compared to the requirement for pelletizing into pellets; therefore, water was added and mixed manually to ensure the equal distribution of moisture throughout the wood flour. Depending on the wood species, the moisture content of the wood flour was maintained between 10 and 15%. Maciejewska et al. [26] mentioned that the wood particles must be brought to the moisture content of 12–17% of weight by volume as required by the pellet press. Under high temperature and pressure, the wood flour was fed into a pellet mill and forced through a round opening called a “die” of quarter-inch thickness where the flour is compacted to form a solid mass of pellets. The high temperature in the machine allows the lignin in the wood to heat and this acts as a binder for the formation of pellets [27]. Van der Waals electrostatic or magnetic forces contribute to the process of pellet formation [28,29]. The wood pellets were then allowed to cool.

### 2.3. Characterization of Mill Residues, Wood Flour, and Wood Pellets

The moisture content of the mill residues, hammer mill grindings and screened wood flour, as well as of the wood pellets, was determined for each species. The American Society of Testing and Materials (ASTM) Standard D4442–20 Standard Test Methods for Direct Moisture Content Measurement of Wood and Wood-Based Materials (ASTM international, West Conshohocken, PA, USA) was followed. Moisture content, according to ASTM D9–20, (ASTM international, West Conshohocken, PA, USA) is the amount of water contained in the wood, usually expressed as a percentage of the mass of the oven-dry wood. The oven-dry method was followed where the samples were kept in the oven for 24 h at 103 ± 2 °∁. The moisture content of the samples was calculated using the following formula:

(1)$$\mathrm{Moisture}\mathrm{Content}(\%)\frac{\mathrm{Original}\;\mathrm{mass}-\mathrm{Oven}\;\mathrm{dry}\;\mathrm{mass}}{\mathrm{Oven}\;\mathrm{dry}\;\mathrm{mass}}\;\times \;100\%$$

Similar to moisture content, the bulk density was also measured for the mill residues, hammer mill grindings, screened wood flour, and wood pellets. The ASTM E873–82 Standard Test Method for Bulk Density of Densified Particulate Biomass Fuels (ASTM International, West Conshohocken, PA, USA) was followed. Bulk density is the mass per unit volume of loose materials, powders and other divided “solids” at specified moisture content levels. A wooden cube of volume 1/4 ft^3^ (7079 cm^3^) was taken and weighed to record its weight. Then, the samples were poured with at least 5 droppings of the box from a height of 150 mm to ensure enough settling of the samples at the bottom, and filled up to a certain line as marked. The formula to determine the bulk density is:(2)$$\mathrm{Bulk}\;\mathrm{Density}\;(\mathrm{kg}/m^{3})=\frac{\mathrm{Weight}\;\mathrm{of}\;\mathrm{cubical}\;\mathrm{box}\;\mathrm{and}\;\mathrm{sample}-\mathrm{Weight}\;\mathrm{of}\;\mathrm{cubical}\;\mathrm{box}}{\mathrm{Volume}\;\mathrm{of}\;\mathrm{cubical}\;\mathrm{box}}$$

A Zeiss NVision 40 Scanning Electron Microscope (SEM) (Carl Zeiss Microscopy, LLC, White Plains, NY, USA) with a capacity of up to 1.2 nm resolution was used to observe the morphology of each mesh size fraction of wood flour for each species. The wood flour samples were sputter-coated with an Au/Pd conductive layer before the SEM observations. The thickness of the Au/ Pd coating was 6 nm. Since the particle size of wood flour is comparatively bigger for observation using SEM images, the magnification of the images was 50× and with a surface area of 100 µm in a high vacuum 3.56 × 10^−6^ Torr; the electron source voltage was 3 kV.

A Ro-Tap Shaker (Retsch Inc., Newton, PA, USA) was used for the particle size analysis of the wood flour. The American National Standards Institute/American Society of Agricultural Engineers (ASAE) S319.4 Method of Determining and Expressing Fineness of Feed Materials by Sieving (ASAE, St. Joseph, MI, USA) was followed. The geometric mean diameter or medium size of particles by the mass, geometric standard deviation of log-normal distribution by mass in ten-based logarithm, and geometric standard deviation of log-normal distribution by mass in natural logarithm were calculated based on the standard.

Similarly, the aspect ratio, which is the ratio of the images’ width and height that describes the particles’ shape, was calculated for each mesh size of each species. The small sample of each fraction/mesh size of the wood particles was placed on the black background with a scale and the image was taken through a digital camera. Image J software was used to express the average aspect ratio. The formula for the calculation of the aspect ratio is:(3)$$\mathrm{Aspect}\;\mathrm{Ratio}\;\mathrm{of}\;\mathrm{wood}\;\mathrm{flour}=\frac{\mathrm{Major}\;\mathrm{axis}}{\mathrm{Minor}\;\mathrm{axis}}=\frac{\mathrm{width}}{\mathrm{height}}$$

The pellet dimensions were determined using a Vernier caliper (Master Gage and Tool Co., Danville, VA, USA). The diameter and length of the pellets were determined. For diameter, two types of measurement, one normal and the other with the angle of 90˚ of the previous measurement, was taken. Five wood pellets for each sample were taken randomly and the average value was calculated. Similarly, to determine the Pellets Durability Index (PDI), ASAE S269.4 (ASAE, St. Joseph, MI, USA) was followed. A durability tester/tumbler (Seedburo Equipment Company, Des Plaines, IL, USA) was used in the experiment and the formula to calculate PDI is:(4)$$\mathrm{PDI}=\frac{\mathrm{Mass}\;\mathrm{of}\;\mathrm{pellets}\;\mathrm{retained}\;\mathrm{on}\;\mathrm{the}\;\frac{1}{8}\;\mathrm{inch}\;\mathrm{sieve}\;\mathrm{after}\;\mathrm{tumbling}}{\mathrm{mass}\;\mathrm{of}\;\mathrm{pellets}\;\mathrm{before}\;\mathrm{tumbling}}\;\times \;100\%$$

To measure the ash content of the wood pellets, a thermo-gravimetric analyzer (TGA) 701 from LECO Corporation (St. Joseph, MI, USA), with an operating voltage of ~230 V, was used. To control the atmosphere inside the furnace, pneumatic gas supply/air of 3.10 bars was set up. The allowable temperature range in the TGA was 25–1000 °C. A maximum of 19 samples could be analyzed per batch. To be used in TGA, the wood pellet samples were powdered and loaded in a crucible. After 24 h, the value of the ash content was recorded on the computer system connected to the machine (1.2x TGA701, LECO Corporation, St. Joseph, MI, USA).

### 2.4. Statistical Analysis

A one-way Analysis of Variance (ANOVA) with a 0.05 significance level was used to determine the significant differences in the means of two or more variables. The statistical association of wood species and mesh size with the bulk density, moisture content and aspect ratio of the wood flour was analyzed. Here, wood species and mesh size are the independent variables whereas bulk density, moisture content and aspect ratio are the dependent variables. Tukey’s test was performed as post-hoc analysis, whenever applicable, to figure out which groups in the sample differ more or less.

## 3. Results and Discussion

Figure 2 shows the SEM micrographs of the 40-mesh size wood flour for each of the four wood species in the study. Images were taken at 50× magnification and a scale bar of 100 µm. SEM images of 18, 35, 60, 80, and 100 mesh sizes of wood flour, for each of the species, are shown in Figure A1 in Appendix A. Compared to Cedar and Maple wood flour, the flour of Pine and Spruce-Fir was finer. Softwoods are flexible, whereas hardwoods are stiffer; furthermore, the chemical compositions between these species can contribute to the differences observed [30]. Despite being a softwood, wood flour fibers of Cedar appeared thicker and less fractured than the wood flour of Pine and Spruce-Fir. The presence of extractives in Cedar could cause some lubricity during processing, thus contributing to the observations of thicker fibers. The wood flour particles appear as layered tube wood-like structures [31,32]. The production of wood flour results in fiber bundles rather than individual fibers [33]. Depending on the species, differences can be observed in the appearance of the flour particles. Maple flour appears with a smoother surface than the other wood species particles. In general, softwood flour fibers are long and thin, which is attributable to the abundance of longitudinal tracheids. Hardwood fibers are short and thick, which is attributable to the abundance of radial and axial cellular components. Because of this, the softwood wood flour appears rougher compared to the maple flour in the SEM images. Even though sieving was performed for a considerable length of time, it was difficult to obtain the diameter of the particles equal to the mesh size [34]. This results in the discrepancies in the diameters of the particles for the same mesh size in the images.

Two parameters of wood fillers addressed in the study of WPC material properties are the wood species and mesh size [35]. Figure 3 is a graphical representation of the relationship between mesh size and moisture content as well as mesh size and bulk density. Compared to other wood species, the moisture content of cedar flour was highest (8.4%), with spruce-fir being the lowest (5.4%), and the bulk density of maple flour was highest (268 kg/m^3^), with pine flour (142 kg/m^3^) being the lowest. In the case of softwoods, the moisture content of the flour is proportional to the moisture of the residue feedstock. Wood moisture content is an important controlling parameter in the manufacturing of WPCs as a moisture level above 1% can cause the composite to foam in the extruder, i.e., produce microvoids of irregular and heterogeneous shapes [36,37].

A one-way ANOVA test was run to determine the association of mesh size with moisture content and bulk density. From the statistical test, it was observed that the type of wood species had an association with the moisture content and bulk density of the wood flour. Similarly, the test showed there is no significant difference in the values of moisture content and bulk density with the change in mesh size of the wood flour. However, for all species, comparatively, the 80-mesh flour had the highest moisture content and 40-mesh the lowest moisture content. Similarly, the bulk density of 40-mesh wood flour was highest and of 80-mesh the lowest for all species. This shows there is an inverse relationship between moisture content and bulk density among different mesh sizes of wood flour. Patterson [38] mentioned that wood filler size between 40 and 80 mesh is the easiest to work with in the manufacturing of WPC products. The abnormality in the behavior of the moisture content and bulk density of wood flour larger and finer than 40 and 80 mesh, respectively, could also be a contributing reason.

Table 1 lists the results of the wood flour particle size distribution analysis. Wood particle size distribution has a great role in the properties of WPCs and the study of particle size distribution on properties has produced different conclusions by different researchers [29]. From Table 1, it is shown that 68% of the particles are within the range of d_gw_ for 16th and 84th percentile. Even though the same screen size of 0.5 mm was used in the hammer mill for grinding the residues, the mean geometric diameter of the wood flour particles was different among the different species. Spruce-fir wood flour had the smallest sized particles, with a geometric mean diameter of 0.18 mm, and pine flour had the largest sized particles (0.25 mm) compared to the other species. The geometric standard deviation of the diameter of pine flour particles was 0.10 mm, which was the lowest among the different wood species. This might be attributable to the utilization of commercial kiln-dried pine residues in the hammermilling process, which were highly uniform and clean. The fineness of the wood flour particles depends somewhat on the raw material and the manufacturing process used [6]. The resulting particle sizes obtained through the hammermill process vary widely, which is attributable to the hammer speed, the extent of wear on the hammer and screen, screen area, air flow, method of discharge, kind of raw material, moisture content of the raw material, etc. [39,40].

Figure 4 is the graph of aspect ratio for the different mesh sizes of wood flour for each species. Aspect ratio is an important characteristic of wood flour that affects the material properties of WPCs. Typically, the aspect ratio falls in the range of 1 to 5 for wood flour [21,41]. In this study, the aspect ratio was in the range of 2 to 3.5. The aspect ratio of commercial pine flour is 3.3 to 4.5 [42]. From a one-way ANOVA, a correlation among wood species and aspect ratio, as well as mesh size with the aspect ratio values, was not apparent. Similar to statistical results, the graph in Figure 4 shows certain variation in the aspect ratio values for different mesh sizes of different wood species. The aspect ratio, to some extent, is dependent on the grinding methods, moisture mass fraction, and wood species [43]. The same authors mentioned that, in general, there is decrease in aspect ratio with the decrease in the particle size of wood. On average, the 18–40 mesh size flour appears to have the highest values of aspect ratio for all species and 100 mesh has the lowest value. From the post-hoc analysis, it was observed that, compared to other mesh sizes, 100 mesh has the greatest association with aspect ratio. This implies that, for all species, 100 mesh size has the smallest value of aspect ratio. However, it should be noted that obtaining pure mesh size flour fractions from the sieving process is an arduous task.

Figure 5 shows how the moisture content and bulk density changes with the processing steps of mill residues into flour and then into pellets. On average, when the residues are ground into wood flour, the moisture content is reduced by 54%, compressing wood flour into pellets reduced moisture content by 52.3%, and overall, when processing residues into pellets, moisture content is reduced by 76.8%. On average, the moisture content of residues was 2.2 times higher than the wood flour, and wood flour had 3.2 times higher moisture content than the pellets. The moisture content of residue feedstock was 7.4 times greater than that of the pellets. This change in moisture content was significant for cedar, contributing a reduction in moisture content of around 94% in the manufacture of pellets. This might be because of the higher extractives and lignin in cedar contributing to higher adhesion and bonding to form pellets. Bardfield and Levi [44] reported a decrease in wood pellet quality when lignin together with extractive content increases above a threshold level of 34%. Chen et al. [45] also observed the increase in bonding and overall pellet strength with the higher percentage of extractives and lignin. Converse to moisture content, the bulk density increases, which is attributable to processing from residues to flour to pellets. On average, when grinding residues into wood flour, the bulk density increased by 119%, converting from flour to pellets, the bulk density increased by 276%, and, on converting residues into pellets, the bulk density increased by 747%. In other words, the bulk density of wood flour was 2.2 times greater than that of residues, that of pellets was 3.8 times greater than that of flour, and then that of pellets was 8.5 times greater than that of the residues. The change in bulk density was significant for pine, contributing an increase of around 1075% in the bulk density of pellets compared to the residue feedstock. This might be attributable to the utilization of fluffy compressed kiln dried commercial residues of pine with a lower bulk density than the other species. Thus, on average, the moisture content of pellets was 3.2 times less and the bulk density 3.8 times greater than for the wood flour. Mani et al. [46] suggested the size of raw materials has an inverse relation in the bulk density of pellets, i.e., smaller particles produce pellets with higher density because of the larger particle surface area, and vice-versa. However, Bergstrom et al. [47] showed almost equal densities of the pellets irrespective of the variation in particles sizes of the raw feedstock materials.

In Table 2, the physical properties of the wood pellets manufactured using 40-mesh wood flour and unsieved wood flour are presented. In most cases, the ash content of hardwood pellets is around three times greater than those manufactured from softwoods. However, we observed that cedar pellets have a greater ash content than the maple pellets, which is attributable to the presence of higher extractives in the heartwood of cedar. Since a quarter-inch die was used in the pelletizing process, the diameter of the pellets is near to the value. The length of the produced pellets had a greater range of sizes. Compared to the normal greater size of raw materials used in the pelletizing process, the particle size of the raw materials in this study was smaller, which might be the main reason influencing the variation in dimensions of the pellets, i.e., basically the length. Samuelsson et al. [48] observed a smaller correlation between the pellet length and the bulk density. However, they observed higher correlation in the effect of moisture content of raw materials to the pellet length. On the other hand, the durability of cedar and pine pellets was highest, which represents higher bonding of the materials in these pellets. The authors of [29,49] reported moisture in the biomass, to a certain limit (<23%), which acts as a binder during pelletization. Cedar and pine particles had higher moisture content than the rest of the two wood species, which might have contributed to higher bonding of the materials.

The images of wood pellets using 40-mesh size wood flour raw material are shown in Figure 6. Each of the scale bar is 3 cm in length. The production process of wood pellets from softwood wood flour was easier. For maple, during the pellet manufacturing from the wood flour, a large number of fine particles were generated in the waste collection chamber relative to the softwoods. Less desirable chemical composition of the particles is one of the important causes behind the higher generation of fines [29]. This might be attributed to the lower lignin content in maple that reduces binding of the particles and induces higher friction during pelletization [50]. Some of the past researchers [45,51] have mentioned that producing pellets from hardwood raw materials is arduous due to higher frictional forces in the compression channels of the die compared to the softwoods, leading to blockage of the pellet mill. On the other hand, researchers such as Thiffault et al. [51] have suggested that, under adequate conditioning and pelletizing, hardwood residues can still be processed into pellets. In our experiment, pellets from maple were manufactured successfully. However, there was significant production of fine particles during the manufacturing process. The type and amount of extractives contributes to the fundamental differences between hardwoods’ and softwoods’ pelletizing properties [52]. Extractive content influences the pelletizing process and pellets’ quality [15,48]. The role of lignin plasticization when the biomass is processing has been studied by several authors. The authors of [53,54] are some of those who suggested that, as the temperature increases, lignin becomes more flexible, thus increasing the flow of molecules. This results in the enhancement of surface contact between particles, enabling the inter-penetration of polymer chain ends and segments between adjacent fibers. Consequently, new secondary bonds and entanglements are established once the polymer is cooled down below the glass-transition temperature of lignin. In a comparative study of the properties of pellets produced using unsieved wood flour and 40-mesh wood flour, it was observed that using the fractionated 40-mesh wood flour produced better quality pellets than the unsieved wood flour. This might be attributable to the greater uniformity of the particle size distribution of the fibers for the fractionated 40-mesh flour. The uniform size of the particles is also responsible for the smooth appearance of the wood pellets, which increases ease of storage and transportation.

## 4. Conclusions

The following conclusions are drawn from the present study:On average, when grinding residues into wood flour, moisture is reduced by 54%, pressing wood flour into pellets reduces moisture by 52.3%, and overall, when processing residues to pellets, moisture content is reduced by 76.8%. A decrease in moisture content is a critical factor from a transportation perspective.On average, the bulk density increased by 119% on the comminution of residues into wood flour, it increased by 276% when converting wood flour to pellets, and, when converting residues into pellets, this value increased, on average, by 747%. An increase in bulk density is an important factor from a transportation point of view.Compressing fine wood flour into pellets produced pellets with a higher range in terms of dimension, and this value varied based on species in the study.It was found that 40-mesh fractionated wood flour produced better quality pellets than the unsieved wood flour.It is challenging to obtain the pure fractionated mesh sizes of the wood flour in terms of dimensions and aspect ratio even after sieving for a longer period of time. Because of the tendency of fibers with different mesh sizes to exhibit different physical properties, it is difficult to find direct relationships between material properties.

## 5. Future Research Work

This study presents a characterization of wood flour and wood pellets, mostly focusing on moisture content and bulk density, with the aim of reducing transportation costs. We expect that both of these raw materials have similar influence on the material properties of WPCs. The second part of this study will involve using these raw materials as fillers in plastics and comparing the physical and mechanical properties of the resulting WPCs. Future research work on processing flour into pellets along with using additives used in WPCs manufacturing, application with different polymers and different formulations, etc. for a better comparative study is highly recommended.

## Figures and Tables

**Figure 1 polymers-13-02487-f001:**
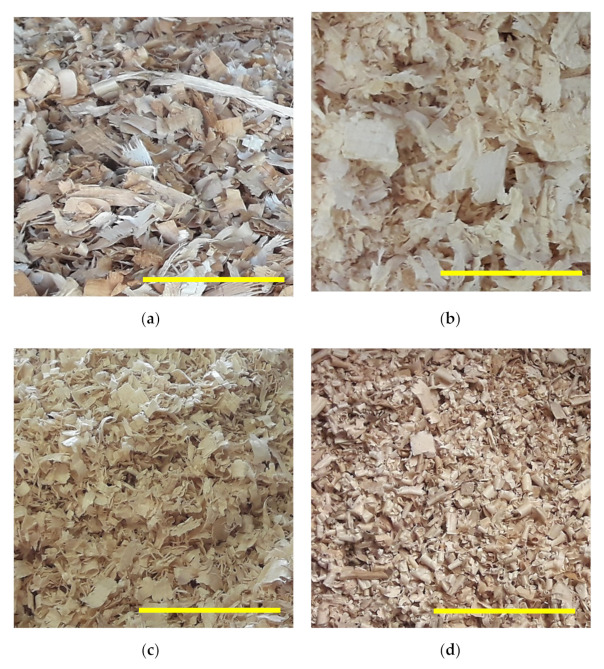
Mill residues of (**a**) White Cedar (**b**) White Pine (**c**) Spruce-Fir (**d**) Red Maple.

**Figure 2 polymers-13-02487-f002:**
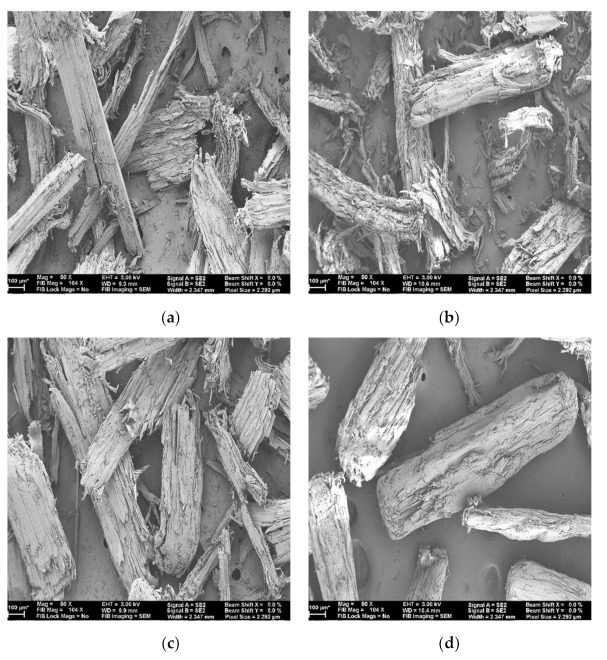
SEM images of wood flour: (**a**) Cedar 40 mesh, (**b**) Pine 40 mesh, (**c**) Spruce-Fir 40 mesh, (**d**) Maple 40 mesh.

**Figure 3 polymers-13-02487-f003:**
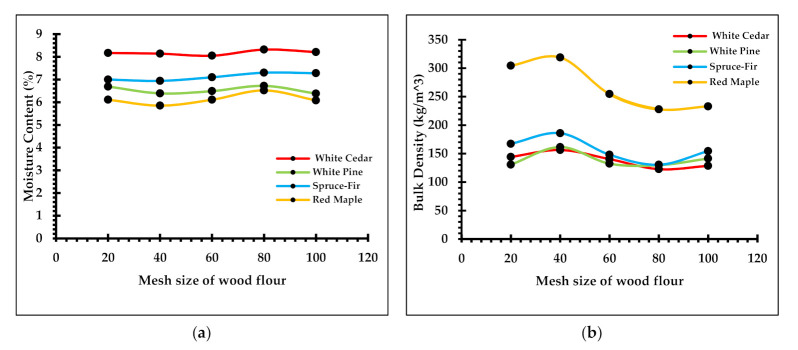
Graphs of (**a**) change in moisture content with mesh size, and (**b**) change in bulk density with mesh size.

**Figure 4 polymers-13-02487-f004:**
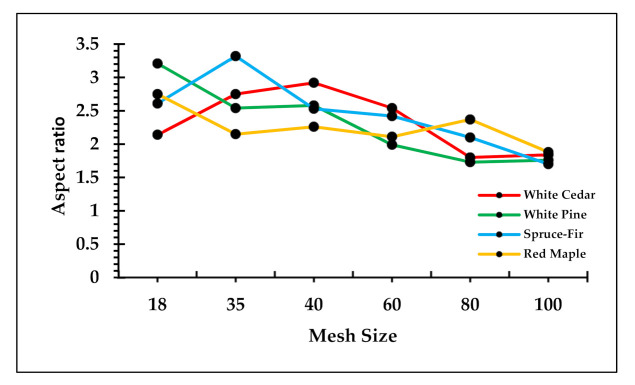
Relation of aspect ratio with different mesh sizes.

**Figure 5 polymers-13-02487-f005:**
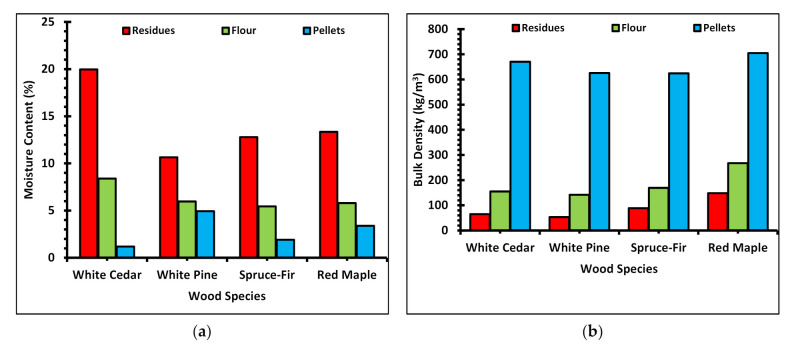
Plots of (**a**) change in moisture content with wood processing steps, and (**b**) change in bulk density with different wood processing steps.

**Figure 6 polymers-13-02487-f006:**
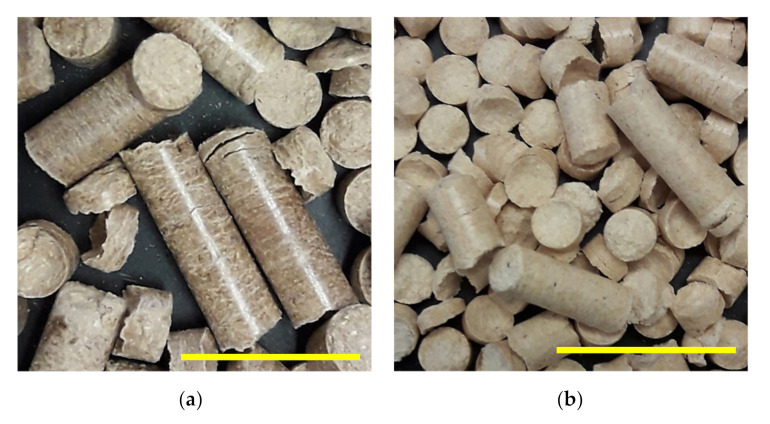
Wood pellets manufactured from 40 mesh wood flour: (**a**) White cedar, (**b**) White pine, (**c**) Spruce-fir, (**d**) Red maple.

**Table 1 polymers-13-02487-t001:** Particles size distribution analysis of wood flour for different wood species.

Species Type	Geometric Mean Diameter (d_gw_) in mm	d_gw_ for 84th Percentile in mm	d_gw_ for 16th Percentile in mm	Geometric Standard Deviation (S_gw_) in mm	Log-Normal Geometric Standard Deviation (S_log_) in mm
White Cedar	0.22	0.03	1.84	0.12	0.23
White Pine	0.25	0.03	2.56	0.10	0.17
Spruce-Fir	0.18	0.02	1.57	0.11	0.26
Red Maple	0.21	0.03	1.52	0.14	0.27

**Table 2 polymers-13-02487-t002:** Various properties of wood pellets pelletized from unsieved and 40-mesh flour.

Wood Species	Moisture Content (%)	Bulk Density (kg/m^3^)	Ash (gm)	Avg. Diameter (inches)	Avg. Length (inches)	Avg. Range of Length (inches)	Durability (%)
White Cedar (unsieved)	1.19	670	0.53	0.24	0.40	0.16–0.67	95
White Cedar (40 mesh)	2.67	699	0.45	0.24	0.40	0.14–0.73	97
White Pine (unsieved)	4.94	625	0.18	0.24	0.35	0.12–0.60	93.8
White Pine (40 mesh)	4.12	634	0.19	0.24	0.40	0.14–0.73	97
Spruce-Fir (unsieved)	1.91	624	0.32	0.24	0.48	0.19–0.93	75.8
Spruce-Fir (40 mesh)	2.57	671	0.28	0.23	0.46	0.17–0.84	82.5
Red Maple (unsieved)	3.39	705	0.41	0.24	0.49	0.17–0.93	67.2
Red Maple (40 mesh)	0.12	738	0.44	0.23	0.48	0.17–0.95	83.8

## Data Availability

All the necessary data are included within the article. Upon request, the data in the article can be made available from the corresponding authors.

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
