# Peer review of "Comparative Study of the Properties of Wood Flour and Wood Pellets Manufactured from Secondary Processing Mill Residues"

_polymers, 2021, doi:10.3390/polym13152487_

Round 1

Reviewer 1 Report

  1. The manuscript is well organized, and the subject is well presented. The methods used are sound and the presentation and discussion of results is logical.
    The manuscript requires some major revisions to bring it to a level worthy of publication. My recommendations are detailed below:
  2. The current study investigates the wood flour of mill residues in the USA, for this the authors test four wood species and manufacture wood flour and wood pellets and compare their performance. The authors found the converting to pellets can help reduce moisture and improve the bulk density.
  3. The abstract needs some improvement, I had to read it twice to understand what was done exactly. Please consider reviewing the abstract and highlight the novelty, major findings and conclusions.
  4. Please remove the section 1.2 Objectives of the study and just keep it within the introduction.
  5. Also at the end of the introduction the authors are encouraged to answer the following question: What is the research gap did you find from the previous researchers in your field? Mention it properly. It will improve the strength of the article.
  6. Figure 1 is it possible to replace it with a closer view using a microscope of the different mill residues? Add a scale bar and take close pictures of them its better for visualising it rather than looking at it in plastic bags, from these photos it all looks the same to me.
  7. Figure 2 is very large, please consider moving some of the SEM images to the appendix or only keep few which are most important to show/deliver your point.
  8. Also all SEM images in Figure 2 are squashed, please make sure to fix this issue.
  9. “Two parameters of wood fillers mostly addressed” remove the word mostly, not sure why you have it in this sentence.
  10. “A one-way ANOVA test was run to determine the association between mesh size and moisture content and mesh size and bulk density.” This sentence needs to be rephrased, which are the input parameters and which are the output parameters?
  11. “Patterson [33] has mentioned” remove has.
  12. Moderate English changes required.
  13. “Typically, the aspect ratio falls in the range of 1 to 5 for wood flour” please reference this information.
  14. “From a one-way ANOVA, a correlation between wood species and aspect ratio values was not apparent.” Why? Error too large or results confound? Please explain and discuss.
  15. “This is also obvious from the graph as different species have different higher or lower values of aspect ratio” rephrase this sentence, it does not read well.
  16. “From the post-hoc analysis, it was observed compared to other mesh sizes” I do not understand this sentence, please check and rephrase.
  17. “In other words, on average the moisture” remove in other words.
  18. “This might be because of the higher extractives and lignin in cedar contributing to higher adhesion and bonding to form pellets” please support this explanation with a reference or relate it to past studies in the open literature similar or closely related to this work.
  19. “. This might be attributable to the utilization of fluffy compressed kiln dried commercial residues of pine with a lower bulk density than the other species.” Same as before, support this justification with reference(s).
  20. All images in Figure 6 are squashed and must be fixed.
  21. The results are merely described and is limited to comparing the experimental observation. The authors are encouraged to include a discussion section and critically discuss the observations from this investigation with existing literature.

Reviewer 2 Report

Review of manuscript "Comparative study of the properties of wood flour and wood pellets manufactured from secondary processing mill residues."

Dear authors, you did a very good job with your manuscript. It is excellently written.

I have only minor suggestions:

Abstract: Please be more specific about the results (some numbers) of your research.

"Because of this, shipping costs often exceed the ...." this is a well-known problem and I am glad authors are dealing with it.

The introduction:

"About 83.4 million tons of primary...." please add newer information than from 2010.

"Wood-plastic composites..." Please add a source, for example here: 

https://doi.org/10.1007/978-981-10-0655-5_2

"Studies on wood flour production are quite ..." I agree with the authors. Please mention some of older sources, for example here:

"There has been considerable research..." You can find at least some particular  research dealing with WPC manufacture using wood pellets, please check for example here: 

https://doi.org/10.1007/s10443-010-9134-2

Materials and methods:

Please mention also one-way ANOVA as a statistical tool.

Results and Discussion:

Page 12: Please discuss more deeply softwood and hardwood pellets production, there are many sources available, for example here:

https://pubs.acs.org/doi/abs/10.1021/ef0503360

Round 2

Reviewer 1 Report

All questions answered, paper can be accepted